# Annexin Induces Cellular Uptake of Extracellular Vesicles and Delays Disease in *Escherichia coli* O157:H7 Infection

**DOI:** 10.3390/microorganisms9061143

**Published:** 2021-05-26

**Authors:** Ashmita Tontanahal, Ida Arvidsson, Diana Karpman

**Affiliations:** Department of Pediatrics, Clinical Sciences Lund, Lund University, 221 85 Lund, Sweden; ashmita.tontanahal@med.lu.se (A.T.); ida.arvidsson@med.lu.se (I.A.)

**Keywords:** Annexin A5, extracellular vesicles, enterohemorrhagic *Escherichia coli*, Shiga toxin, hemolytic uremic syndrome, phagocytes

## Abstract

Enterohemorrhagic *Escherichia coli* secrete Shiga toxin and lead to hemolytic uremic syndrome. Patients have high levels of circulating prothrombotic extracellular vesicles (EVs) that expose phosphatidylserine and tissue factor and transfer Shiga toxin from the circulation into the kidney. Annexin A5 (AnxA5) binds to phosphatidylserine, affecting membrane dynamics. This study investigated the effect of anxA5 on EV uptake by human and murine phagocytes and used a mouse model of EHEC infection to study the effect of anxA5 on disease and systemic EV levels. EVs derived from human whole blood or HeLa cells were more readily taken up by THP-1 cells or RAW264.7 cells when the EVs were coated with anxA5. EVs from HeLa cells incubated with RAW264.7 cells induced phosphatidylserine exposure on the cells, suggesting a mechanism by which anxA5-coated EVs can bind to phagocytes before uptake. Mice treated with anxA5 for six days after inoculation with *E. coli* O157:H7 showed a dose-dependent delay in the development of clinical disease. Treated mice had lower levels of EVs in the circulation. In the presence of anxA5, EVs are taken up by phagocytes and their systemic levels are lower, and, as EVs transfer Shiga toxin to the kidney, this could postpone disease development.

## 1. Introduction

Annexins are a group of highly conserved proteins that interact with phospholipids on cellular membranes in a calcium-dependent manner [1,2]. They are both intracellular and extracellular [3] and, by forming a scaffold on membranes, exert a multitude of functions affecting membrane dynamics and the cytoskeleton, stabilizing membrane defects [4], inducing membrane repair [5] and linking membranes [6]. The latter can enable contact between cells [7] or extracellular vesicles (EVs) [8]. Furthermore, by forming a connection between the membrane and the intracellular environment, annexins mediate signal transduction, vesicle budding, endocytosis [9] and exocytosis [10]. Endogenous annexins are also involved in phagocytosis by interacting with actin in the cytoskeleton [11].

The annexins differ mostly at their N-terminal and thereby exhibit varying characteristics. Annexins exhibit potent anti-inflammatory, anticoagulant and fibrinolytic properties [1]. Annexin A5 (anxA5 or annexin V) was the first member of this family to be crystallized [12] and has been extensively studied. AnxA5 stabilizes membranes and induces membrane repair [5]. This is enabled by its binding to phosphatidylserine (PS) and self-assembly to form a 2D-lattice on the cell surface in a calcium-dependent manner [4]. AnxA5 has also been shown to mediate membrane fusion [13]. Its binding to PS enables it to be used as a diagnostic tool for detection of apoptotic cells [14] and EVs [15,16]. 

EVs include exosomes, shed microvesicles and apoptotic bodies and contain components of the parent cell such as receptors, proteins, RNAs and lipids [17,18]. During their formation, the physiological asymmetry of the plasma membrane is disrupted and phospholipids, such as PS, that are normally present on the inner leaflet of the lipid bilayer plasma membrane, flip to the outer membrane.

Circulating EVs are of particular interest in thrombotic conditions, because EVs have prothrombotic properties, such as exposure of tissue factor and PS [19]. One such condition in which the role of EVs has been studied is hemolytic uremic syndrome (HUS). HUS is often associated with gastrointestinal infection caused by enterohemorrhagic *Escherichia coli* (EHEC). EHEC are non-invasive bacteria that colonize the large intestine without causing bacteremia [20]. During EHEC infection, the bacteria release the unique bacteriophage-encoded virulence factor Shiga toxin [21]. Infected patients develop diarrhea or hemorrhagic colitis and in severe cases may develop HUS characterized by acute kidney injury, thrombocytopenia and microangiopathic hemolytic anemia [22]. EVs play an important role in this condition as they sequester Shiga toxin, thereby allowing its transfer from the gut to the kidney, where it is taken up within EVs, released, and leads to renal cell death [23]. Patients with EHEC-associated HUS have elevated levels of PS-positive [19] and Shiga toxin-positive blood cell-derived EVs [23]. Similarly, EHEC-infected mice develop renal and hematological disease [24], and have high levels of circulating blood cell-derived EVs [23]. Renal damage in mice is characterized by glomerular and tubular cell injury, elevated blood urea nitrogen, indicating poor kidney function, and fibrinogen deposition in glomeruli [24,25]. As in human disease, murine EVs transfer Shiga toxin from the circulation to the kidney [23].

Due to the ability of anxA5 to link membranes and to readily bind PS on EVs, we hypothesized that it may affect the disease process in EHEC infection by binding to EVs and interfering with Shiga toxin transfer. The aim of this study was to investigate the effect of anxA5 on EVs in the presence of phagocytes in vitro and to further study its effect on EVs during EHEC infection in vivo. To this end, we used anxA5-coated EVs derived from either human whole blood or HeLa cells and investigated their uptake by human or murine monocyte/macrophage cell lines. An established mouse model of oral EHEC infection was used to study the effects of anxA5 on mouse survival and circulating microvesicle levels.

## 2. Methods

### 2.1. Cell Cultures

THP-1 cells (human monocyte, Sigma-Aldrich, Steinheim, Germany) were cultured in RPMI 1640 medium (Gibco, Waltham, MA, USA) and differentiated to macrophages as described below. HeLa cells (a kind gift from L. Johannes, Institute Curie, Paris, France) were cultured in Dulbecco’s Modified Eagle Medium (DMEM, Gibco). RAW264.7 cells (murine monocyte, Sigma-Aldrich) were cultured in DMEM supplemented with glutamine (2 mM, Thermo Fisher Scientific, Waltham, MA, USA). All cells were cultured in medium supplemented with 10% fetal bovine serum and 1% penicillin-streptomycin (both from Gibco) and grown in 5% CO_2_ at 37 °C.

### 2.2. Generation of Fluorescent Blood Cell-Derived Extracellular Vesicles

Human whole blood was drawn from healthy adult male volunteers (*n* = 2, not using any medications) into citrated blood collection tubes (Becton Dickinson, Franklin Lanes, NJ, USA) diluted 1:1 with FluoroBrite DMEM (Gibco) and centrifuged at 2500× *g* for 10 min at room temperature (RT). Plasma was removed and the blood cells were washed with phosphate buffer saline (PBS) without calcium (HyClone, Logan, UT, USA) and centrifuged at 2500× *g* for 10 min at RT. Washed blood cell membranes were stained with PKH26 Red Fluorescent Cell Linker Kit (PKH26, Sigma-Aldrich) according to the manufacturer’s protocol and washed with FluoroBrite DMEM. PKH26-stained blood cells (diluted 1:10 in FlouroBrite DMEM) were stimulated with or without Shiga toxin 2 (Stx2, 200 ng/mL, Phoenix Lab, Tufts Medical Center, Boston, MA, USA) for 30 min, followed by calcium ionophore A23187 (5 μM, Sigma-Aldrich) for 40 min at 37 °C under gentle rocking. Blood cells were separated by centrifugation at 10,000× *g* for 10 min at RT, twice. Supernatants containing PKH26-stained EVs were centrifuged at 20,000× *g* for 40 min at 4 °C to obtain an EV-rich suspension which was washed and concentrated by differential centrifugation twice at 20,000× *g* for 40 min at 4 °C. These EVs were used for detection of uptake by differentiated THP-1 cells (Table 1).

The use of blood from healthy volunteers was approved by the Regional Ethics review board of Lund University with the written informed consent of the subjects.

### 2.3. Generation of Fluorescent HeLa Cell-Derived EVs

HeLa cells (2 × 10^6^ cells) were seeded out in a T75 culture flask (Thermo Fisher Scientific) 24 h before the start of the experiment. Cells were washed twice with Hank’s Balanced Salt Solution (Gibco) and stained with PKH26 as above. Cells were stimulated with calcium ionophore A23187 as above. PKH26-stained EVs were centrifuged at 20,000× *g* for 40 min at 4 °C to obtain an EV-rich suspension, which was washed and concentrated from the HeLa cell culture media by differential centrifugation three times at 20,000× *g* for 40 min at 4 °C. These EVs were used for detection of uptake and for the detection of PS on RAW264.7 cells described below (Table 1).

### 2.4. Extracellular Vesicle Uptake

RAW264.7 cells (60,000 cells/well) were seeded out 24 h prior to the start of experiment in a μ-Slide 8-well glass bottom (Ibidi, Gräfelfing, Germany). THP-1 cells (30,000 cells/well) were similarly seeded out and treated with phorbol-12-myristate-13-acetate (200 nM, Sigma-Aldrich) for 72 h before the start of experiment in order to differentiate into macrophages. Cells were washed twice with FluoroBrite DMEM. Both cells and EVs, described above, were pre-incubated with or without recombinant His-tagged human anxA5 (10 μg/mL, Annexin V, BioVision Inc., Milpitas, CA, USA) for 30 min at 37 °C. Differentiated THP-1 cells were then incubated with blood cell-derived EVs (shed after stimulation of blood cells with or without Stx2), while RAW264.7 cells were incubated with HeLa cell-derived EVs, for 2 h at 37 °C.

After incubation, cells were washed with FluoroBrite DMEM and stained with PKH67 Green Fluorescent Cell Linker Kit for membrane labelling (Sigma-Aldrich) according to the manufacturer’s protocol. Cells were fixed with paraformaldehyde (2%, Histolab Products, Askim, Sweden) for 30 min in the dark and washed twice with FluoroBrite DMEM. Cell nuclei were stained with HCD Nuclear mask blue stain (Thermo Fisher Scientific) for 30 min in the dark and washed twice with FluoroBrite DMEM. Cells were visualized in a Ti-E inverted flourescence microscope equipped with a Nikon structured illumination microscopy module (Nikon Instruments Inc., Tokyo, Japan) and imaged using NIS elements AR software v.5.11.01.

### 2.5. Quantification of Extracellular Vesicle Uptake by Cells

Images of differentiated THP-1 cells and RAW264.7 cells stained green (PKH67), that had been incubated with EVs stained red (PKH26), were captured using a fluorescence microscope. Image stacks at 40× magnification were converted to maximal intensity images. Stained cells were outlined with a threshold above the background to select the area occupied by cells. Data are presented as the EVs/cell (fold increase), the quantified area being 1.4 mm^2^ of individual wells and an average of two images/well; two wells per condition were quantified using Fiji ImageJ software v2.1.0/1.53c (NIH, Bethesda).

### 2.6. Phosphatidylserine Exposure on RAW264.7 Cells

Experiments were designed to determine if RAW264.7 cells expose more PS in the presence of EVs. RAW264.7 cells (60,000 cells/well) were seeded out in an μ-Slide 8-well glass bottom (Ibidi) 24 h before the start of experiment. PKH26-stained HeLa cell-derived EVs were isolated, as described above, and incubated with the cells for 1 h. Certain cells were left untreated in FlouroBrite DMEM for the same time points at 37 °C. Calcium ionophore A23187 5 μM was used as the positive control or calcium ionophore A23187 5 μM with EDTA (10 mM, Sigma-Aldrich), as the negative control, both for 5 min at 37 °C. After incubation, cells were stained with NucBlue Live Cell Stain ReadyProbes reagent (Invitrogen, Carlsbad, CA, USA) and anxA5:FITC (3 μL/well, BD Biosciences, Franklin Lakes, NJ, USA), according to the manufacturers’ protocols. Cells were visualized using the Ti-E inverted fluorescence microscope equipped with a Nikon structured illumination microscopy module.

### 2.7. Quantification of Phosphatidylserine Expression on Cells

Exposure of PS on cells was defined as a measure of anxA5-FITC binding. Images of RAW264.7 cells, that had been incubated with EVs stained red (PKH26), were captured and the background subtracted. Nuclei (stained blue) were defined using a threshold above the background in order to select stained nuclei and quantified. The threshold for FITC (green) fluorescence was determined based on untreated samples (sample without EVs). AnxA5-FITC binding was assessed as a measure of fluorescence, i.e., (mean green fluorescence intensity x area occupied by the green fluorescence) per cell in 1.4 mm^2^ area of individual wells (one well per condition) using Fiji ImageJ software v2.1.0/1.53c.

### 2.8. Mice

BALB/c mice were bred in the animal facility of the Centre for Comparative Medicine, Medical Faculty, Lund University. Both male and female mice aged 8–12 weeks were used in the experiments.

Institutional Review Board statement: All animal experiments were approved by the Animal ethics committee of Lund University in accordance with the guidelines of the Swedish National Board of Agriculture and the European Union directive for the protection of animals used for scientific research. Approval number: M148-16 (date of approval 2017-01-25) and 5.8.18-17452/20 (date of approval 2020-12-16).

### 2.9. Bacterial Strain

*Escherichia coli* O157:H7 strain 86–24 [26] producing Stx2 was used in all animal experiments. The strain was previously genetically and phenotypically characterized [27] and a spontaneously obtained streptomycin-resistant derivative was used [24].

### 2.10. Escherichia coli O157:H7 Infection Protocol in Mice

Mice received streptomycin sulphate salt (5 g/L, Sigma-Aldrich) in drinking water for 24 h before inoculation throughout the experiment, in order to reduce the normal gut microflora at colonization. Mice were subjected to fasting for food and not water for 16 h prior to inoculation. Thereafter, mice were anesthetized with isoflurane (Forene, Abbott, Wiesbaden, Germany), and *E. coli* O157:H7 bacterial suspension was inoculated intragastrically (100 µL of bacteria was suspended in 20% sucrose and 10% sodium bicarbonate vehicle (Sigma-Aldrich) in MiliQ water to achieve a final concentration of 10^9^ CFU/mL or 100 µL vehicle alone, using a soft polyethylene catheter (Clay Adams, Parsippany, NJ, USA) that was withdrawn immediately afterwards. Post inoculation food was reintroduced ad libitum.

Animals were monitored two to three times daily. Weight was recorded daily and expressed as percent body weight compared to the initial weight on day -1 before fasting. Appearance of clinical signs such as ruffled fur, lethargy, hunched posture, decreased activity or ≥20% weight loss were considered end points for euthanasia, as previously described [28]. In this mouse model, mice develop disease starting on day 4–5. In separate experiments, mice were sacrificed on day 3, before the development of clinical disease or on day 5, when one PBS-treated *E. coli* O157:H7-infected mouse developed clinical disease or at the end of the experiment, on days 7 or 9 after inoculation, at which point even unaffected mice were sacrificed.

Fecal samples were collected on days 1, 3 and 5, weighed and dissolved in PBS. The samples were serially diluted and 100 µL were plated overnight on a Luria Bertani-agar plate containing streptomycin (50 µg/mL). Colonies were counted and the serotype confirmed using an *E. coli* O157 latex kit (Oxoid, Basingstoke, UK).

### 2.11. Detection of Phosphatidylserine-Positive Murine Extracellular Vesicles

Murine plasma samples from mice sacrificed on day 5 after *E. coli* O157:H7 infection were diluted 400 times (final volume 100 μL) in filtered FluoroBrite DMEM and stained with anxA5:FITC (1:100) or with EDTA (10 mM), as a negative control, for 30 min at RT in the dark. Staining was terminated by addition of 200 μL filtered FluoroBrite DMEM. Flow cytometry was performed using Amnis^®^ CellStream^®^ (Luminex, Austin, TX, USA). All buffers were filtered through a 0.2 μm pore-size filter (Pall Corporation, Ann Arbor, MI, USA) to remove particles and aggregates. Samples were acquired using Small Particle Mode (flow rate = 3.66 μL/min with a threshold of 1000 on all channels), 10 μL of each sample were measured with FSC, SSC and 488 laser power set to 100%. Data were analyzed using CellStream^®^ analysis software (Luminex).

### 2.12. Treatment with anxA5

BALB/c mice were injected intraperitoneally with anxA5 (100 or 500 µg/kg, BioVision) diluted in sterile PBS (100 µL, vehicle) or vehicle alone, every day for up to 6 days from the day of *E. coli* O157:H7 inoculation. Certain mice were sacrificed on day 3 or 5, i.e., before the end of the full length of the experiment on day 7 or 9, when all mice in the infected group developed symptoms.

### 2.13. Blood Collection

Blood was collected in citrate (0.9%, Sigma-Aldrich) via a heart puncture under isoflurane anesthesia. Samples were centrifuged at 1500× *g* for 15 min and 13,000× *g* for 3 min at RT and stored at −80 °C, until analyzed. These samples were used to measure blood urea nitrogen levels, quantify anxA5 concentration and EVs in plasma.

All mice were tested for bacteremia using blood culture flasks (Becton Dickinson) and were found to be negative.

### 2.14. Blood Urea Nitrogen Measurement

Plasma blood urea nitrogen was measured using a QuantiChrom Urea assay kit (BioAssay systems, Hayward, CA, USA) according to the manufacturer’s instructions.

### 2.15. Quantification of anxA5 Concentration in Mouse Plasma

AnxA5 levels in mouse plasma were measured using ZYMUTEST™ Annexin V (HYPHEN Biomed, Neuville-sur-Oise, France) according to the manufacturer’s instructions and analyzed using a GloMax Discover System (Promega, Madison, WI, USA).

### 2.16. Quantification of Platelet-Derived Murine EVs

Murine plasma samples were diluted 50 times in filtered PBS without Ca (HyClone) and stained with rat anti-mouse CD41-AlexaFluro488 or rat IgG1-AlexaFluro488 (negative control, 1:100, Bio-Rad Laboratories, Hercules, CA, USA) for 30 min at RT in the dark. Staining was terminated by addition of PBS without Ca (200 μL, HyClone). All buffers were filtered through a 0.2 μm pore-size filter. Flow cytometry was performed as described above.

### 2.17. Immunofluorescence Staining of Murine Kidney

Mice were sacrificed and tissue was collected, fixed in 4% paraformaldehyde (Histolab, Gothenburg, Sweden) and embedded in paraffin. Tissue sections were stained as previously described [24], and images were obtained using the Ti-E inverted fluorescence microscope and analyzed using Fiji ImageJ software.

Tissues were analyzed in blinded fashion for fibrinogen deposition in glomeruli, and the intensity of staining was graded using a scoring system of no staining (0), low (1+), medium (2+) or high (3+) intensity (Appendix A). Entire kidney sections were visualized at 10× magnification until all glomeruli were counted (ranging from 118–219). A degree of fibrinogen staining intensity was assigned to each glomerulus. The degree of fibrinogen staining intensity (0−1−2−3) was multiplied by the number of glomeruli with a specific fibrinogen staining intensity and the total level of intensity in each kidney section was thereby calculated. Immunofluorescence staining was performed on *E. coli* O157:H7-infected mice treated with anxA5 (500 μg/kg) or PBS vehicle and uninfected mice as healthy controls.

### 2.18. Stx2 Detection in Mouse Kidneys

Stx2 detection in the kidneys of *E. coli* O157:H7-infected mice sacrificed on day 5 was performed by immunofluorescence staining, Stx2-ELISA or mass spectrometry. Tissue was collected upon sacrifice, and fixed as above. For immunofluorescence staining, tissue sections were stained with camelid anti-Stx1 and Stx2 (List Biological Laboratories, Campbell, CA, USA) followed by goat anti-llama:FITC (Invitrogen). For Stx2 ELISA and mass spectrometry, proteins were extracted and processed from paraffin-embedded kidney sections as previously described [29]. Stx2 ELISA was carried out using camelid anti-Stx1 and Stx2 as the capture antibody followed by mouse anti-Stx2B (BEI Resources, Manassas, VA, USA). Details are given in the Appendix A.

### 2.19. Interaction of anxA5 with Stx2

A direct interaction between anxA5 and Stx2 was assessed. An activated Sequi-Blot PVDF membrane (Bio-Rad Laboratories, Hercules, CA, USA) was coated with anxA5 (1 or 10 μg) or Stx2 (positive control, 1 μg, Phoenix Lab) or bovine serum albumin (negative control, 1% BSA, Sigma-Aldrich) and allowed to dry. The PVDF membrane was blocked with 1% BSA for 1 h at RT. The membrane was washed three times with 0.05% PBS-Tween (PBS-T, Medicago, Uppsala, Sweden) and incubated with Stx2 diluted in 1% BSA (200 ng/mL) for 1 h at RT. After washing, the membrane was incubated with mouse anti-Stx2 (1:100, Santa Cruz Biotechnology, Dallas, TX) for 1 h at RT, washed and further incubated with polyclonal goat anti-mouse immunoglobulin HRP (1:1000, Dako, Glostrup, Denmark) for 1 h at RT. The membrane was washed and developed using Pierce ECL Plus Western Blotting Substrate (Thermo Fisher Scientific) according to manufacturer’s protocol and visualized using a GelDoc Touch Imaging System (Bio-Rad Laboratories).

### 2.20. Interaction of anxA5 with E. coli O157 Lipopolysaccharide

To assess a possible interaction between anxA5 and *E. coli* O157 lipopolysaccharide (O157LPS), a nitrocellulose membrane was incubated with anxA5 (positive control, 0.5 μg) or O157LPS (10 or 20 μg, a gift from R. Johnson, Public Health Agency, Guelph, ON, Canada) or 1% BSA (negative control) for 1 h at RT. The membrane was washed three times with 0.05% PBS-Tween and blocked with 1% BSA for 1 h at RT. The membrane was then incubated with anxA5 (2 μg/mL) in HEPES buffered saline (Gibco) overnight at 4 °C. The membrane was washed and incubated with mouse anti-His tag (1:5000) for 1 h at RT, washed and further incubated with anti-mouse HRP (1:1000, Dako). The membrane was washed, developed and visualized as above.

### 2.21. Cytotoxicity/Viability Assay

HeLa cells (10,000 cells/well) were incubated with purified Stx2 (1–100 ng/mL) with or without anxA5 (1, 5 and 10 μg/mL) diluted in serum-free DMEM for 24 h in 5% CO_2_ at 37 °C. Cytotoxicity of Stx2 was assessed using Alamar Blue (Thermo Fisher Scientific) according to the manufacturer’s protocol and measured using the Glomax Discover System.

### 2.22. Statistical Analysis

Differences between groups were assessed by the two-tailed Mann–Whitney test or by the Kruskal–Wallis multiple-comparison test when comparing more than two groups, followed by Dunn’s procedure for comparisons between specific groups. Kaplan–Meier survival curves were analyzed using the log-rank test. A *p*-value ≤ 0.05 was considered significant. All statistical analyses were calculated using Prism 9 version 9.0.2 (GraphPad, La Jolla, CA, USA).

## 3. Results

### 3.1. AnxA5 Increases Cellular Uptake of Extracellular Vesicles

Blood cell-derived EVs were incubated with THP-1 cells demonstrating EV uptake by the cells. The uptake of EVs increased when the vesicles were treated with anxA5 compared to untreated vesicles (Figure 1a,b), but not when only the cells were treated with anxA5 but not the vesicles. A similar result was obtained with blood cell-derived EVs, from Stx2-stimulated cells, incubated with THP-1 cells (Appendix A). Likewise, HeLa cell-derived EVs incubated with RAW264.7 cells exhibited an increase in EV uptake when the EVs were pretreated with anxA5 (Figure 1c), but not when only the cells were pretreated. Taken together, anxA5 treatment of the EVs, from blood cells or HeLa cells, accounted for the enhanced EV uptake, while anxA5 treatment of the recipient THP-1 or RAW264.7 cells had no effect on EV uptake.

### 3.2. Extracellular Vesicles Induce Excess anxA5 Binding to RAW264.7 Cells

Phosphatidylserine expression on cells induces anxA5 binding [30], and we therefore examined if EVs induced PS expression on RAW264.7 cells. EVs isolated from calcium ionophore-stimulated HeLa cells were incubated with RAW264.7 cells for 1 h and PS exposure on the cells quantified by binding of anxA5:FITC. In these experiments, the EVs and the cells were not treated with anxA5. In the presence of EVs, a significant increase in anxA5:FITC binding was observed in comparison to cells not treated with EVs (Figure 2), indicating that the EVs induced PS exposure on the cells.

### 3.3. Murine Extracellular Vesicles Express Phosphatidylserine

Levels of circulating PS-positive EVs in mouse plasma were analyzed in samples taken on day 5 after inoculation with *E. coli* O157:H7. Three of a total of 7 mice exhibited increased PS-positive EVs on day 5 before the development of clinical disease (Figure 3). These data indicate that murine EVs expose PS on their surface.

### 3.4. AnxA5 Treatment Delayed Disease Symptoms in Murine E. coli O157:H7 Infection

The effect of anxA5 treatment was investigated in mice infected with *E. coli* O157:H7. Mice inoculated with *E. coli* O157:H7 (*n* = 8) developed symptoms starting on day 4 after inoculation (Figure 4a,b). Mice inoculated with *E. coli* O157:H7 and treated with the lower concentration of anxA5 (100 μg/kg) developed symptoms on day 6–7 (Figure 4a) and mice treated with the higher concentration (500 μg/kg) developed symptoms on day 6–9 after inoculation (Figure 4b). Weight loss was observed in all mice inoculated with *E. coli* O157:H7 (Figure 4c,d), and there was no difference in bacterial colonization between mice treated with anxA5 or vehicle (Figure 4e,f). The results suggest that treatment with anxA5 delayed disease symptoms in the infected mice.

In a separate set of experiments, mice were sacrificed on day 3, before the development of clinical disease or on day 5, when one PBS-treated *E. coli* O157:H7-infected mouse developed clinical disease, to obtain samples at a specific time point. These included *E. coli* O157:H7-infected mice that were treated with the PBS vehicle (*n* = 18) or anxA5 (500 μg/kg, *n* = 18) and uninfected mice treated with the PBS vehicle (*n* = 6) or anxA5 (500 μg/kg, *n* = 8). No difference in weight or bacterial colonization (in infected mice) was observed between these groups (data not shown).

### 3.5. Blood Urea Nitrogen Levels in E. coli O157:H7-Infected Mice

Blood urea nitrogen (BUN) was measured in mouse plasma taken from mice on day 3 as described above. No difference in the BUN levels was observed between *E. coli* O157:H7-infected mice and uninfected mice, anxA5-treated or untreated (Figure 5a). Samples were also taken on day 5. The one vehicle-treated *E. coli* O157:H7-infected mouse that developed clinical disease exhibited high levels of BUN in plasma in comparison to the other *E. coli* O157:H7-infected mice that were sacrificed before they developed clinical disease (Figure 5b). In general, infected mice had slightly higher BUN levels even before the development of disease, but there was no difference between anxA5-treated and untreated mice. At the end of the experiment when all *E. coli* O157:H7-infected mice, anxA5-treated or untreated, developed clinical disease, high levels of BUN were observed in all mice compared to the vehicle-treated and anxA5-treated uninfected mice (Figure 5c).

### 3.6. AnxA5 Detection in Mouse Plasma

AnxA5 concentration was analyzed in mouse plasma samples taken on day 3 after inoculation with *E. coli* O157:H7. A higher anxA5 concentration was found in *E. coli* O157:H7-infected mice treated with anxA5 (Figure 6a) when compared to infected PBS vehicle-treated mice. Likewise, a significant difference in anxA5 levels was observed between the anxA5-treated and vehicle-treated uninfected mice (Figure 6b).

### 3.7. AnxA5 Lowered Circulating Platelet-Derived Extracellular Vesicles

In vitro data suggested that anxA5 on EVs induces their uptake by phagocytic cells (Figure 1). In vivo, this should correspond to reduced levels of circulating EVs. Levels of circulating platelet-derived vesicles in mouse plasma were analyzed in samples taken on day 3 and 5 after inoculation with *E. coli* O157:H7. A significant decrease in circulating platelet-derived vesicles was observed between the anxA5-treated and vehicle-treated *E. coli* O157:H7-infected mice (Figure 7), suggesting that anxA5 treatment lowered EV levels and possibly promoted clearance of circulating vesicles.

### 3.8. Fibrinogen Deposition in Murine Kidneys

Once mice developed fulminant kidney disease during *E. coli* O157:H7 infection enhanced fibrinogen deposition in the kidneys has been demonstrated [24]. This effect was seen here as well; mouse kidneys exhibited fibrinogen deposition with no difference between those treated with anxA5 or left untreated (data not shown). The effect of anxA5 on fibrinogen deposition in the kidneys of infected and uninfected mice was therefore investigated in mice sacrificed on day 5 after inoculation with *E. coli* O157:H7. Infected mice were treated with the PBS vehicle (*n* = 5, one developed disease) or anxA5 (500 μg/kg, *n* = 5, no disease in these mice) and uninfected mice were treated with the PBS vehicle (*n* = 4) or anxA5 (500 μg/kg, *n* = 3) and sacrificed. There was no difference in glomerular fibrinogen deposition in infected and control mice (Appendix A) at this stage of infection.

### 3.9. Stx2 Was Not Detectable in Murine Kidneys

Stx2 was not detected in the kidney of mice infected with *E. coli* O157:H7 by immunofluorescence staining, Stx2 ELISA or by mass spectrometry (data not shown).

### 3.10. AnxA5 Did Not Bind to Stx2 or O157LPS and Did Not Protect HeLa Cells from Stx2-Induced Cytotoxicity

AnxA5 treatment delayed symptoms in *E. coli* O157:H7-infected mice, and we therefore assessed if anxA5 binds directly to Stx2 or O157LPS. The Stx2 sample showed a strong signal (positive control). A signal was not detected for binding between Stx2 and anxA5 or BSA, thus anxA5 did not bind to Stx2 (data not shown). Similarly, the interaction between anxA5 and O157LPS was investigated. AnxA5-His showed a strong signal (positive control). No signal was detected for O157LPS or BSA, indicating that anxA5 did not bind to O157LPS (data not shown).

The effect of Stx2-induced cytotoxicity in the presence of anxA5 in vitro was assessed. HeLa cells were incubated with varying concentrations of purified Stx2 with or without anxA5 for 24 h. AnxA5 did not exert a protective effect (Figure 8).

## 4. Discussion

Extracellular vesicles promote the development of renal failure associated with EHEC infection by exposing a prothrombotic surface on their membranes, by participating in hemolysis and, most importantly, by transferring the main virulence factor Stx2 to the kidney [23,31]. Here, we show that treatment of EVs with anxA5 induced their uptake by human and murine phagocytes and decreased their plasma levels in vivo. This could lower the toxin load in the circulation and explain the delay in symptoms observed in anxA5-treated EHEC-infected mice. As anxA5 did not affect bacterial colonization of the gut, or neutralize the effects of Stx2, the bacteria would continuously release Stx2 in the gut which, after binding and uptake by blood cells, is released into the circulation within EVs [22,32]. Thus, the removal of EVs by phagocytes would postpone disease until a critical amount of toxin accumulates in the kidney. This effect seems to be dose-dependent, as a higher concentration of anxA5 was more protective. We therefore propose that anxA5 can induce the uptake of EVs by phagocytes and thereby eliminate them from the systemic circulation.

Uptake of EVs by phagocytes was associated with the presence of anxA5 on their surface, while the presence of anxA5 on the surface of the phagocytic cells themselves did not contribute to this process. Furthermore, the EVs induced exposure of PS on the phagocyte membranes, thereby explaining the mechanism by which anxA5-coated EVs would bind to phagocytes, as PS is the phospholipid ligand preferred by annexins [1], particularly anxA5 [33]. Previous studies have shown that annexins modulate membrane flow and induce membrane linking and fusion in the presence of calcium (reviewed in [34]). Annexins collaborate with soluble N-ethylmaleimide-sensitive factor attachment protein receptors (SNARE proteins) in late endocytic pathways and membrane fusion, leading to exocytosis [35,36]. Furthermore, anxA2 was shown to promote clathrin-dependent endocytosis of stress-associated EVs [37], and anxA5 promoted particle phagocytosis by the retinal pigment epithelium [38]. Thus, we propose that exogenous anxA5 on EVs can contribute to alterations in the EV membranes, making them, under certain conditions, more susceptible to phagocytic uptake.

Previous studies have addressed the interaction of EVs with cells in the presence of anxA5. AnxA5 did not affect the binding of PS-positive EVs from monocytic THP-1 cells to platelets but inhibited membrane fusion [39]. On the other hand, anxA5 inhibited the uptake of EVs from hypoxia-induced stem cells by human umbilical cord endothelial cells (HUVECs) [40], whereas it did not affect the uptake of alveolar macrophage-derived EVs by alveolar epithelial cells [41]. Presumably, the variability in these data depend on discrepant experimental conditions affecting cells, EVs and the microenvironment they are exposed to. PS on apoptotic cells is considered to be a phagocytic signal, and although anxA1 and anxA2 were found to enhance phagocytosis, anxA5 inhibited phagocytosis of dying cells at higher concentrations, most probably by formation of a lattice that covered PS and shielded the cell membrane (reviewed in [42]). In contrast, polymerization of anxA5 on cell surfaces alters membrane curvature, eliciting an endocytic pinocytic pathway [43]. It is, however, unknown how the interaction between anxA5 on EVs and PS on recipient cells impacts the cell membrane, and thereby uptake of EVs by cells. Thus, data regarding the ability of anxA5 to enhance or inhibit phagocytosis of EVs are as yet inconclusive and most probably dependent on the conditions used. In the in vivo model, we could show that anxA5 decreased circulating EVs and delayed disease, suggesting that harmful EVs were eliminated in the presence of anxA5.

AnxA5 may have exerted several protective effects in *E. coli* O157:H7-infected mice by reducing prothrombotic PS-positive EVs and by delaying Stx2 transfer to the kidney within the EVs. PS on EV membranes serves as a procoagulant surface on which the prothrombinase complex is assembled, consisting of prothrombin and coagulation factors Va and Xa, and the intrinsic tenase complex, consisting of factors VIIIa and IXa, activating factor X and subsequently leading to thrombin formation [44]. In addition, the extrinsic pathway of coagulation is also activated on PS, as tissue factor binds factor VIIa on PS-positive membranes [39,44]. Our studies in patients that develop HUS after EHEC infection showed high levels of PS and tissue factor-positive EVs in the circulation [19]. Using the murine model of EHEC infection, we can show herein that mice also have PS-positive EVs in the circulation, even before the development of clinical disease. AnxA5 was shown to reduce tissue factor on the surface of dying cells by inducing its cell entry [45], which may be a protective antithrombotic mechanism.

In our EHEC infection mouse model, we have shown excessive fibrinogen deposition in the glomeruli of sick mice [24]. The same was true in the current study; however, once mice developed fulminant disease, fibrinogen deposition in glomeruli was comparable, regardless of anxA5 treatment. We therefore investigated fibrinogen deposition before the development of disease but could not detect a difference between anxA5-treated and untreated mice at an early stage of infection. Similarly, the blood urea nitrogen levels were high in anxA5-treated and untreated mice during fulminant disease but were not elevated before clinical disease developed. We also attempted to detect Stx2 in renal tissue by immunofluorescence, ELISA or by mass spectrometry, but toxin levels during EHEC infection are much too low to enable detection in tissue, as the LD50 for Stx2 was found to be 5 × 10^−5^ mg/kg body weight [46]. Thus, we can assume that reduced levels of circulating EVs decreased the delivery of pro-thrombotic and toxin-positive EVs to the kidney and thereby delayed the onset of disease, but could not quantify toxin in the kidney.

Treatment with anxA5 may exert additional beneficial effects in the host infected with *E. coli* O157:H7, than those described above. AnxA5 has anti-inflammatory properties. In an ApoE^(−/−)^ mouse model of vascular inflammation, it reduced leukocyte recruitment and systemic inflammation as measured by levels of pro-inflammatory mediators [47]. Likewise, anxA5 reduced inflammation in a mouse model of endotoxemia by reducing cytokines [48] and blocking binding of lipopolysaccharide to toll-like receptor 4 [49]. Importantly, anxA5 has been registered in a clinical trial for the treatment of COVID infection (https://ichgcp.net/clinical-trials-registry/NCT04748757) or in healthy individuals (clinicaltrials.gov/ct2/show/study/NCT04850339). AnxA5 may thereby exert multiple protective effects during infection, particularly EHEC infection, reducing circulatory harmful prothrombotic toxin-positive EVs and decreasing the inflammatory response to infection. A delay in development of HUS after prodromal EHEC-associated gastrointestinal infection occurs may enable the attenuation of the course of disease by appropriate hydration and volume expansion, thereby preventing acute kidney injury [50]. As we could show that anxA5 delayed disease in the EHEC mouse model, we suggest that anxA5 be investigated as a potential treatment for this infection.

## Figures and Tables

**Figure 1 microorganisms-09-01143-f001:**
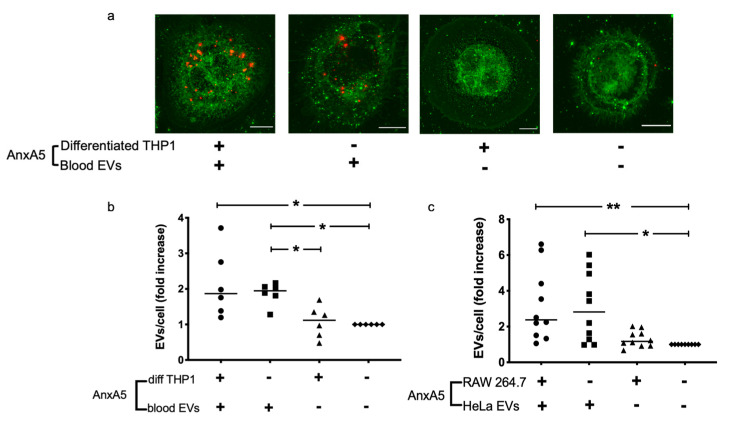
Effect of anxA5 on extracellular vesicle uptake by THP-1 and RAW 264.7 cells. (**a**) Stained differentiated THP-1 cells (green dye) and stained blood cell-derived extracellular vesicles (EVs, red dye) were pre-incubated with or without anxA5, after which the cells and EVs were co-incubated. Left panel: Both cells and EVs that were pre-incubated with anxA5 showed localization of EVs (red) within the cell (green), indicating that anxA5 induced an increase in EV uptake. 2nd to left panel: Only EVs were pre-incubated with anxA5 before co-incubation, with cells showing localization of EVs (red) within the cell (green), indicating that anxA5-induced an increase in EV uptake. 2nd to right panel: Cells, but not EVs, were pre-incubated with anxA5 before cells were co-incubated with EVs showing low or no localization of EVs within the cell, indicating low or no EV uptake. Right panel: Cells and EVs were not pre-incubated with anxA5 and exhibited low or no localization of EVs within the cell, indicating low or no EV uptake. All the panels were acquired using a structure illumination microscope at 100× magnification. Stained EVs that have not been taken up by the cells are not visible in the background as the EV suspension was washed away. Scale bar: 10 μm. (**b**) Differentiated THP-1 cells and blood cell-derived extracellular vesicles (EVs) were pre-incubated with or without anxA5 and co-incubated. The presence of anxA5, both on EVs and cells, resulted in a significant increase in EV uptake by differentiated THP-1 cells in comparison to the absence of anxA5, both on EVs and cells. The presence of anxA5 only on EVs also resulted in a significant increase in EV uptake by differentiated THP-1 cells when compared to the presence of anxA5 only on cells or the absence of anxA5, both on EVs and cells. (**c**) RAW 264.7 cells and HeLa cell-derived EVs were pre-incubated with or without anxA5 and were then co-incubated. The presence of anxA5, both on EVs and cells, resulted in a significant increase in EV uptake by RAW 246.7 cells in comparison to the absence of anxA5 both on EVs and cells. The presence of anxA5 only on EVs also resulted in a significant increase in EV uptake by RAW 246.7 cells when compared to the absence of anxA5, both on EVs and cells. EV uptake by cells is presented relative to the sample in which neither cells nor EVs were pre-incubated with anxA5 (defined as 1) in order to account for experimental variability. The bar indicates the median EVs/cell (fold increase). *: *p* < 0.05, **: *p* < 0.01, Kruskal–Wallis multiple-comparison test followed by Dunn’s procedure.

**Figure 2 microorganisms-09-01143-f002:**
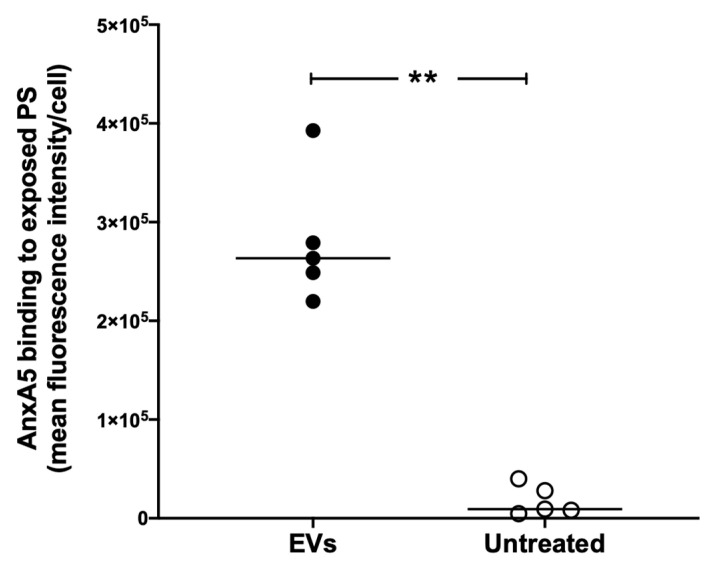
EVs induced exposure of phosphatidylserine on RAW264.7 cells. Incubation of HeLa cell-derived EVs with RAW 264.7 cells resulted in a significant increase in phosphatidylserine exposure on the cells in comparison to untreated RAW 264.7 cells (without EVs). The bar represents median of mean fluorescence intensity/cell. **: *p* < 0.01, two-tailed Mann–Whitney test.

**Figure 3 microorganisms-09-01143-f003:**
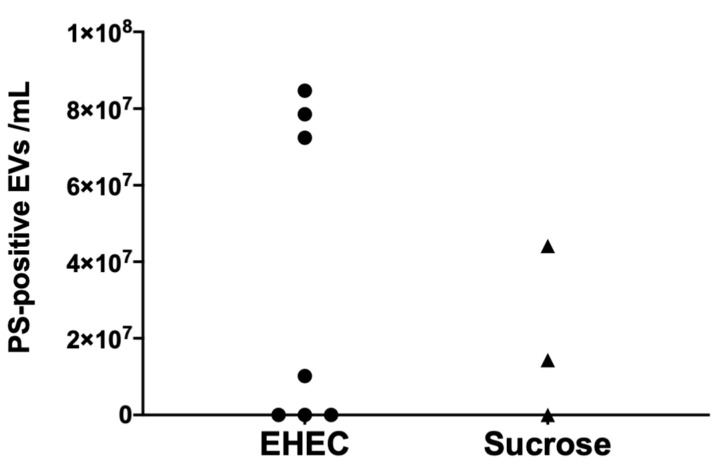
Murine extracellular vesicles express phosphatidylserine. Plasma from *E. coli* O157:H7-infected mice (*n* = 7, one of these mice was sick) or uninfected controls (*n* = 3) sacrificed on day 5 post *E. coli* O157:H7 inoculation exhibited circulating phosphatidylserine-positive extracellular vesicles (EVs). PS: phosphatidylserine.

**Figure 4 microorganisms-09-01143-f004:**
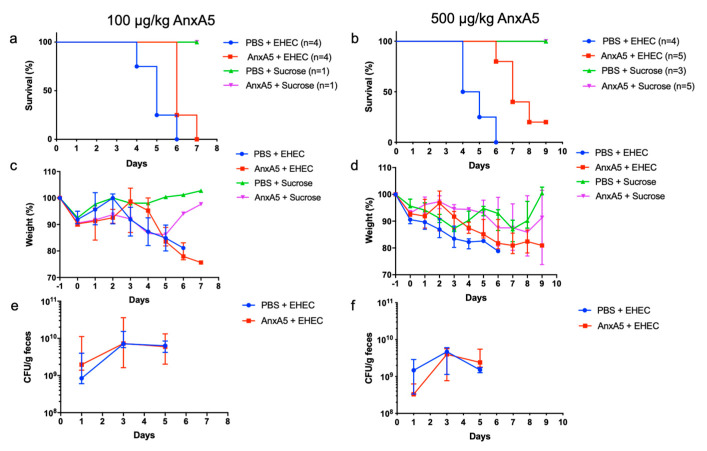
Survival, weight and fecal bacterial counts in *E. coli* O157:H7-infected mice. BALB/c mice were treated with anxA5, or PBS-vehicle, from the day of *E. coli* O157:H7 inoculation and daily for 6 days. (**a**) Survival in *E. coli* O157:H7-infected mice treated with PBS (blue line) or anxA5 (100 μg/kg, red line) and uninfected mice treated with PBS (green line) or anxA5 (100 μg/kg, purple line). Mice in the untreated group developed symptoms from day 4 post-inoculation, while mice in the anxA5-treated group started to develop symptoms on day 6. (**b**) Survival in *E. coli* O157:H7-infected mice treated with PBS (blue line), anxA5 (500 μg/kg, red line) and uninfected mice treated with PBS (green line) or anxA5 (500 μg/kg, purple line). Mice in the untreated group started to develop symptoms on day 4 post-inoculation, while the mice in the anxA5-treated group started to develop symptoms on day 6 or later. (**c**) Weight changes in mice treated with anxA5 (100 μg/kg) or left untreated starting 1 day before inoculation, at the start of fasting, until day 7, by which time all mice in the infected groups developed symptoms. (**d**) Weight changes in mice treated with anxA5 (500 μg/kg) or left untreated, starting 1 day before inoculation, at the start of fasting, until day 9, when the experiment ended. (**e**) Bacterial colony-forming units in feces of *E. coli* O157:H7-infected mice on day 1, 3, 5 showing no difference in colonization between *E. coli* O157:H7-infected mice treated with PBS (blue line) and anxA5 (100 μg/kg, red line). (**f**) Bacterial colony forming units in feces of *E. coli* O157:H7-infected mice on day 1, 3, 5 showing no difference in colonization between *E. coli* O157:H7-infected mice treated with PBS (blue line) and anxA5 (500 μg/kg, red line). Data presented as median and range.

**Figure 5 microorganisms-09-01143-f005:**
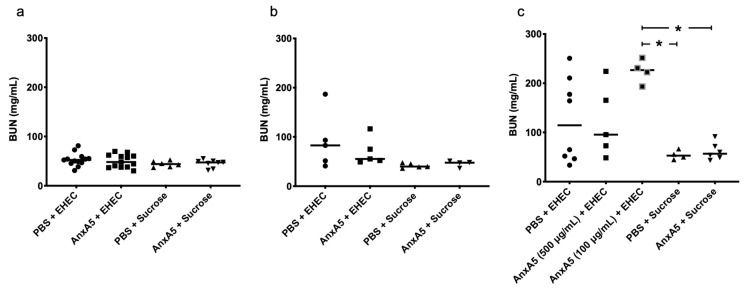
Blood urea nitrogen in *E. coli O157:H7*-infected and control mice. (**a**) Blood urea nitrogen (BUN) levels in mouse plasma of *E. coli* O157:H7-infected mice treated with PBS (*n* = 13) or anxA5 500 μg/kg (*n* = 13) and uninfected mice treated with PBS (*n* = 6) or anxA5 500 μg/kg (*n* = 8) sacrificed on day 3, showing no difference in BUN levels. (**b**) BUN levels in mouse plasma of *E. coli* O157:H7-infected mice treated with PBS (*n* = 5) or anxA5 500 μg/kg (*n* = 5) and uninfected mice treated with PBS (*n* = 5) or anxA5 500 μg/kg (*n* = 4) sacrificed on day 5 when one of the PBS-treated infected mice developed clinical disease. (**c**) BUN levels in mouse plasma of *E. coli* O157:H7-infected mice treated with PBS (*n* = 8) or anxA5 500 μg/kg (*n* = 5) or anxA5 100 μg/kg (*n* = 4) and uninfected mice treated with PBS (*n* = 4) or anxA5 (*n* = 6, both concentrations of anxA5 have been combined as the concentration of anxA5 had no effect on BUN levels in uninfected mice) sacrificed at the end of the experiment. *E. coli* O157:H7-infected mice showed higher levels of BUN compared to the uninfected mice. The bar represents the median. *: *p* < 0.05, Kruskal–Wallis multiple-comparison test followed by Dunn’s procedure.

**Figure 6 microorganisms-09-01143-f006:**
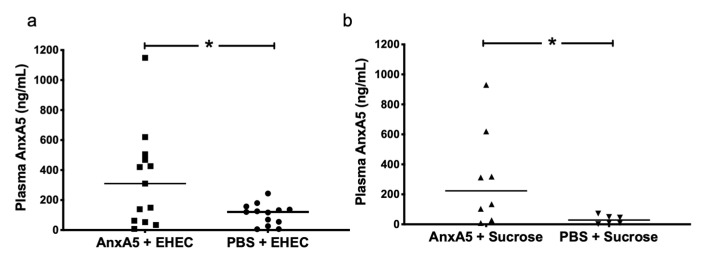
AnxA5 plasma concentrations in *E. coli* O157:H7-infected and control mice. (**a**) AnxA5 concentrations in the plasma of *E. coli* O157:H7-infected mice treated with PBS (*n* = 13) or anxA5 500 μg/kg (*n* = 13) and sacrificed on day 3, showing significantly higher anxA5 levels in the treated mice. (**b**) AnxA5 concentrations in the plasma of uninfected mice treated with PBS (*n* = 6) or anxA5 (*n* = 8), showing significantly higher anxA5 levels in treated mice. The bar represents the median. *: *p* < 0.05, two-tailed Mann–Whitney test.

**Figure 7 microorganisms-09-01143-f007:**
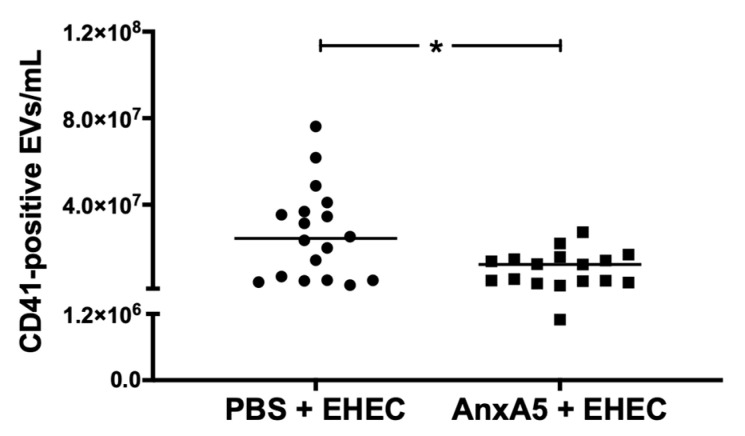
AnxA5 treatment reduced circulating platelet-derived extracellular vesicles. Plasma from *E. coli* O157:H7-infected mice treated with anxA5 (500 μg/kg, *n* = 18) for 3 or 5 days post inoculation had significantly lower numbers of circulating platelet-derived extracellular vesicles in comparison to the PBS (vehicle, *n* = 18)-treated *E. coli* O157:H7-infected mice. The bar represents the median, one outlier has been eliminated. *: *p* < 0.05, two-tailed Mann–Whitney test.

**Figure 8 microorganisms-09-01143-f008:**
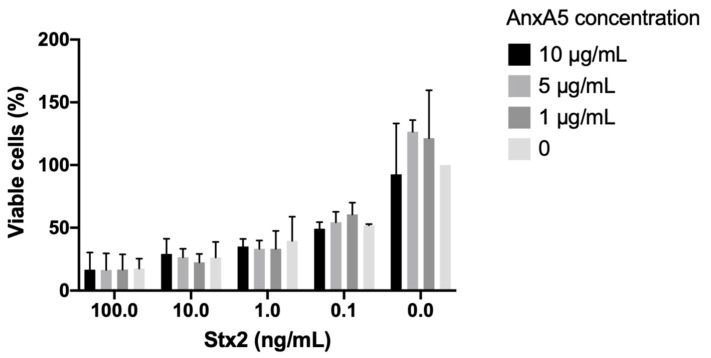
The effect of anxA5 on Stx2-induced cytotoxicity in HeLa cells. Co-incubation of anxA5 and Stx2 with HeLa cells, for 24 h, did not result in a protective effect regarding Stx2-induced cytotoxicity. Serum-free DMEM-treated cells were defined as having 100% viability (column on the right). Data are presented as the median and range of three wells from three separate experiments. No statistical significance was found.

**Table 1 microorganisms-09-01143-t001:** Extracellular vesicles and cells used in in vitro experiments.

EVs/Cells	THP1	RAW 264.7
Whole blood	+ ^a^	-
HeLa ^b^	-	+

a, Stx2 and calcium ionophore stimulated whole blood-derived EVs were used to investigate EV uptake by differentiated THP1 cells in the presence or absence of anxA5. b, HeLa cells were also used for investigating Stx2 cytotoxicity in the presence of anxA5.

## Data Availability

Data are available from the corresponding author upon request.

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
