# Peer review of "Annexin Induces Cellular Uptake of Extracellular Vesicles and Delays Disease in Escherichia coli O157:H7 Infection"

_microorganisms, 2021, doi:10.3390/microorganisms9061143_

Round 1
Reviewer 1 Report
In this paper, studies on effects of annexin A5 (annexin 5) on uptake of extracellular vesicles in the light of infection with enterohemorrhagic Escherichia coli (EHEC) in cellular and animal models are described. The work has been well-conducted, with appropriate controls. Although the results are not spectacular, this is a solid work, expanding our knowledge on the role of extracellular vesicles in EHEC infections.
Some minor points and suggestions:
- Please, use the full name of the species (Escherichia coli) rather than abbreviation (E. coli) in the title.
- Section 1 (Introduction) should contain somewhat more information about Shiga toxin and EHEC, including a brief note that these toxins are encoded by bacteriophages, and what are factors inducing Shiga toxin production. A few literature citations should also be included in such a paragraph.
- More details are required in descriptions of some experimental procedures. For example, was any gradient used for differential centrifugations and what fractions were collected after these procedures (sections 2.2. and 2.3).
- Line 305. The second part of the sentence "and not anxA5 treatment of the recipient cells" is not clear. What is the relationship between the first and second parts of this sentence? Please, rewrite.
- Reading this paper, it is evident that Discussion starts from line 465. However, a heading for such a chapter (section 4 ?) is missing. In fact, a word "Discussion" appears in the legend to Figure 8, without any connection to this legend. Perhaps this is a formatting error, but it should be corrected.
Author Response
In this paper, studies on effects of annexin A5 (annexin 5) on uptake of extracellular vesicles in the light of infection with enterohemorrhagic Escherichia coli (EHEC) in cellular and animal models are described. The work has been well-conducted, with appropriate controls. Although the results are not spectacular, this is a solid work, expanding our knowledge on the role of extracellular vesicles in EHEC infections.
Some minor points and suggestions:
- Please, use the full name of the species (Escherichia coli) rather than abbreviation (E. coli) in the title. RESPONSE: This was corrected.
- Section 1 (Introduction) should contain somewhat more information about Shiga toxin and EHEC, including a brief note that these toxins are encoded by bacteriophages, and what are factors inducing Shiga toxin production. A few literature citations should also be included in such a paragraph. RESPONSE: This was corrected and two references were added.
- More details are required in descriptions of some experimental procedures. For example, was any gradient used for differential centrifugations and what fractions were collected after these procedures (sections 2.2. and 2.3). RESPONSE: This was corrected.
- Line 305. The second part of the sentence "and not anxA5 treatment of the recipient cells" is not clear. What is the relationship between the first and second parts of this sentence? Please, rewrite. RESPONSE: The sentence was rephrased.
- Reading this paper, it is evident that Discussion starts from line 465. However, a heading for such a chapter (section 4 ?) is missing. In fact, a word "Discussion" appears in the legend to Figure 8, without any connection to this legend. Perhaps this is a formatting error, but it should be corrected. RESPONSE: This was a formatting error.
Reviewer 2 Report
In this manuscript, the authors demonstrated that the treatment of Annexin A5 protein could delay the death symptom induced by the infection of human version EHEC in mouse model, which is potentially associated with the Annexin 5A induced phagocyte uptake of Shiga toxin-bound extracellular vesicles and thereby reduces the transfer of toxins to kidney. The experiments were designed and performed well, the manuscript was organized well as well. And the concept itself is very interesting and the findings may benefit the field a lot. I would like to suggest to move this manuscript forward to publication process now. I also have some questions have nothing to do with the revision but we can discuss here:
- Can you detect EHEC-induced diarrhea and kidney damage in your mouse model?
- It is good to know the ratio of EV-bound and free Shiga toxins in the bloodstream after infection in your mouse model.
- If EV is the major transfer route as you discussed in your PLoS Pathogen paper, how those toxins entry targeted kidney cells?
- How to explain the Gb3 KO mice are “super” resistant to Shiga toxins, if the Gb3-independent uptake contributes a lot?
- Can Annexin 5A induced absorption causes unexpected side-effects?
- Do you believe the same mechanism is also happened in human?
Author Response
In this manuscript, the authors demonstrated that the treatment of Annexin A5 protein could delay the death symptom induced by the infection of human version EHEC in mouse model, which is potentially associated with the Annexin 5A induced phagocyte uptake of Shiga toxin-bound extracellular vesicles and thereby reduces the transfer of toxins to kidney. The experiments were designed and performed well, the manuscript was organized well as well. And the concept itself is very interesting and the findings may benefit the field a lot. I would like to suggest to move this manuscript forward to publication process now. I also have some questions have nothing to do with the revision but we can discuss here:
- Can you detect EHEC-induced diarrhea and kidney damage in your mouse model? RESPONSE: The mice do not develop diarrhea but they do develop renal injury. This was added in the introduction, 2nd to last paragraph.
- It is good to know the ratio of EV-bound and free Shiga toxins in the bloodstream after infection in your mouse model. RESPONSE: Free Shiga toxin is almost impossible to measure in humans (in which blood supply is higher) and therefore not possible to measure in mouse plasma.
- If EV is the major transfer route as you discussed in your PLoS Pathogen paper, how those toxins entry targeted kidney cells? RESPONSE: EVs are taken up by cells by various mechanisms, endocytosis, phagocytosis, pinocytosis or membrane fusion. We described this is PMID 28736435. The exact mechanism by which Shiga toxin-positive EVs are taken up is unclear but our PLoS Pathogen paper, with electron microscopy images, would indicate that the EVs are taken up into early endosomes.
- How to explain the Gb3 KO mice are “super” resistant to Shiga toxins, if the Gb3-independent uptake contributes a lot? RESPONSE: EVs can be taken up by Gb3-negative cells but Shiga toxin cannot exert a cytotoxic effect in the absence of Gb3, please see PMID 32523894.
- Can Annexin 5A induced absorption causes unexpected side-effects? RESPONSE: All drugs can cause side-effects, that is why the clinical trials we cite will be important to follow, one in healthy individuals in order to determine if annexin A5 is tolerated well.
- Do you believe the same mechanism is also happened in human? RESPONSE: yes we do as we used human blood cells in in vitro studies but this requires more studies.